# Safety and Efficacy of Extracorporeal Photopheresis for Acute and Chronic Graft-versus-Host Disease

**DOI:** 10.3390/ph17101279

**Published:** 2024-09-27

**Authors:** Eleni Gavriilaki, Eleni Papchianou, Giorgos Karavalakis, Ioannis Batsis, Alkistis Panteliadou, Andriana Lazaridou, Despina Mallouri, Varnavas Constantinou, Paraskevi Karvouni, Paschalis Evangelidis, Anna Papakonstantinou, Apostolia Papalexandri, Panayotis Kaloyannidis, Nikolaos Spyridis, Zoi Bousiou, Anna Vardi, Evangelia Yannaki, Damianos Sotiropoulos, Ioanna Sakellari

**Affiliations:** 1Hematology Department and Bone Marrow Transplant (BMT) Unit, G. Papanicolaou Hospital, 57010 Thessaloniki, Greece; elenipaphianou@gmail.com (E.P.); giorgos.karavalakis@gmail.com (G.K.); iobats@yahoo.gr (I.B.); kirapanteliadou@gmail.com (A.P.); andrianalazaridou@gmail.com (A.L.); vcskg@gmail.com (V.C.); lila.papalexandri@gmail.com (A.P.); pkaloyannidis@yahoo.gr (P.K.); spyridisnik@hotmail.com (N.S.); boussiou_z@hotmail.com (Z.B.); anna_vardi@yahoo.com (A.V.); eyannaki@washington.edu (E.Y.); dsotiro@otenet.gr (D.S.); ioannamarilena@gmail.com (I.S.); 2Second Propedeutic Department of Internal Medicine, Hippocration Hospital, Aristotle University of Thessaloniki, 54642 Thessaloniki, Greece; pascevan@auth.gr; 3Medical School, Aristotle University of Thessaloniki, 54636 Thessaloniki, Greece; dmallouri@gmail.com (D.M.); paraskeu98@gmail.com (P.K.); annapapak86@gmail.com (A.P.)

**Keywords:** allogeneic, acute graft-versus-host disease, chronic graft-versus-host disease, extracorporeal photopheresis, hematopoietic stem cell transplantation

## Abstract

**Background/Objectives:** Despite novel biological agents, steroid-dependent or -refractory graft-versus-host disease (GvHD) remains a severe complication of allogeneic hematopoietic cell transplantation (allo-HCT). Extracorporeal photopheresis (ECP) is an alternative, non-immunosuppressive treatment for patients with acute (aGvHD) or chronic (cGvHD) GvHD. The aim of this study was to investigate the safety and efficacy of ECP in the treatment of acute and chronic GvHD; **Methods:** We prospectively studied 112 patients with cGvHD who received one or more previous lines of treatment and 28 patients with steroid-dependent or refractory grade II-IV aGvHD post-alloHSCT. **Results:** In terms of severe aGvHD, most of the patients (19/28) responded to ECP treatment, while the five-year overall survival (OS) was 34%. After adjustment for several confounder factors, the reduction in immunosuppression (*p* = 0.026) and number of ECP sessions (*p* < 0.001) were associated with improved OS. Regarding chronic GvHD, only 19 patients failed to respond to ECP treatment; though significantly lower rates of response were presented in patients with visceral involvement (*p* = 0.037) and earlier post-transplant GVHD diagnosis (*p* = 0.001). Over a follow-up period of 45.2 [interquartile range (IQR): 5.6–345.1] months, the 5-year cumulative incidence (CI) of cGvHD-related mortality was 21.2% and was significantly reduced in patients with ECP response (*p* < 0.001), while the 5-year OS was 65.3%. **Conclusions:** Our results confirm the safety and efficacy of ECP in patients with GvHD and provide sufficient data for further investigation and the best combination drugs needed such that GvHD will not be the major barrier of allo-HCT in the near future.

## 1. Introduction

Allogeneic hematopoietic cell transplantation (allo-HCT) remains the only potentially curative therapy for various malignant and benign hematological disorders, while the field continues to expand to more indications and more older patients. However, allo-HCT recipients experience several complications such as graft-versus-host disease (GvHD), endothelial injury syndromes, thrombotic events, and infections [1]. Specifically, GvHD is a severe and common complication increasing the morbidity and mortality of these patients [2]. GvHD can either be acute (aGvHD) which is mainly characterized by inflammation of the skin, the liver, and the gastrointestinal tract, or chronic (cGvHD) which can affect almost every organ [3]. According to the latest recommendations of the European Society for Blood and Marrow Transplantation (EBMT), patients undergoing matched related or unrelated donor alloHSCT should receive prophylaxis with a calcineurin inhibitor in combination with an antimetabolite agent [4]. The calcineurin inhibitor can either be tacrolimus or ciclosporin with similar outcomes, while the suggested antimetabolite for patients receiving a myeloablative conditioning (MAC) regimen is methotrexate (MTX) [5,6]. Mycophenolate mofetil can be used for patients receiving reduced intensity or toxicity conditioning regimen regimens [6]. Antithymocyte globulin (ATG) is also recommended in patients undergoing matched unrelated donor alloHSCT [7]. On the other hand, various agents used for GvHD prophylaxis and treatment, such as post-transplantation cyclophosphamide, calcineurin and mammalian target of rapamycin (mTOR) inhibitors have been associated with the development of endothelial injury syndromes, such as HSCT-associated thrombotic microangiopathy (HSCT-TMA) [8,9]. Moreover, GvHD has been recognized as a major risk factor for HSCT-TMA occurrence, while an overlap between these clinical entities has been described [10].

Regarding the initiation of treatment for aGvHD, the decision is based mainly on the clinical manifestations of the patients, although the confirmation of the diagnosis with a biopsy before initiation is also recommended [4]. Treatment is indicated for grade II or higher aGvHD, given the increased risk of infections that are associated [11]. The first-line treatment is corticosteroids (methylprednisolone or prednisone) [12]. On the other hand, the decision for treatment initiation in patients with cGvHD is based on a wide range of criteria such as type and severity of symptoms, disease risk or chimerism. First-line treatment is also prednisone, but in severe cGvHD, the addition of another immunosuppressive agent is suggested [13]. Alternative treatment strategies are usually applied when patients are refractory, dependent, or intolerant to corticosteroids. It is critical for both aGvHD and cGvHD that response to corticosteroids would be assessed early in 1–2 weeks, so the second agent would be initiated. Among a variety of second-line treatments, over 25 years now, extracorporeal photopheresis (ECP) has emerged among one of the first choices [14]. One of the main benefits of ECP is that it is not an immunosuppressive treatment and does not affect the graft-vs-leukemia effect [15]. In ECP, the patient’s mononuclear cells are collected via an apheresis procedure, treated with methoxsalen, exposed to ultraviolet light, and finally returned to the patient [16]. The exact mechanism of action is not yet fully understood, although it seems that ECP does not cause direct cytotoxicity to the alloreactive T cells. On the contrary, it leads to the DNA strands of the treated cells crosslinking, and therefore apoptosis is caused [17,18]. It has been shown that ECP therapy induces immune tolerance through the induction of regulatory T cells [19]. Moreover, the steroid-sparing advantage of ECP has been identified in several studies [20,21].

Given the morbidity that GvHD patients experience, it essential is to consider the investigation of ECP as a treatment approach for allo-HCT recipients who develop GvHD. Thus, the aim of this study is the assessment of the safety and efficacy of ECP, as an early second-line treatment in patients with steroid-dependent or refractory aGvHD, as well as a third-line treatment for patients with cGvHD, in our center.

## 2. Results

### 2.1. ECP for Patients with aGvHD

#### 2.1.1. Patients Characteristics

Our study included 28 patients with aGvHD (12 males and 16 females), with a mean age of 44.3 ± 15 years. A MAC regimen was administered to 16 patients (57.1%), while 8 patients (28.6%) were administered a reduced-toxicity conditioning regimen and 4 (14.3%) a reduced-intensity conditioning regimen. Regarding donor type, 4 patients (14.3%) were transplanted from a sibling donor, 11 patients (39.3%) from a matched unrelated donor, 12 patients (42.9%) from one locus mismatched volunteer unrelated donor, and 1 patient (3.6%) from a haploidentical donor. The disease risk index was very high in 1 patient (3.6%), high in 11 patients (39.3%), intermediate in 14 patients (50%), and low in 2 patients (7.1%). These data are presented in Table 1.

#### 2.1.2. Clinical Characteristics and Treatment Patterns of aGvHD

Acute GvHD was observed at day +17 [Interquartile range (IQR): 8–50], Table 1. There were no overlaps in ECP treatment between aGvHD and cGvHD. Patients treated with ECP due to aGVHD are included only in this group, and not in cGVHD. According to aGvHD manifestations, 9 patients (32.1%) experienced skin, intestine, and liver involvement, skin and intestine involvement was evident in 13 patients (46.4%), while only skin involvement was apparent in 6 patients (21.4%) (see Figure 1). All patients received first-line treatment with corticosteroids with 13 patients (46.4%) being steroid-dependent and 15 patients (53.6%) being steroid-refractory, as shown in Table 1.

#### 2.1.3. Safety and Efficacy of ECP as Second-Line Treatment for aGvHD in a Population Critically Ill and Hospitalized

ECP was commenced at median day +18 post-allo-HCT (IQR: 8–56), while the median number of ECP sessions was 15 (IQR: 4–20). Most patients (19/28) responded to ECP treatment. More specifically, 7 patients (25%) had a partial response, 11 patients (39.3%) had a very good response, and 1 patient (3.6%) had a complete response to ECP (Figure 1). Over a median follow-up period of 9.9 (IQR: 1.7–113) months, immunosuppression was reduced in 12 patients (42.9%) and stopped in 1 patient (3.6%).

Clinically significant bacterial infections were found in 19 patients (67.9%), fungal in 3 patients (10.7%), CMV and EBV reactivation in 19 (67.9%) and 12 (42.9%) patients, respectively, and other viral infections in 6 patients (21.4%). No life-threating infections were observed.

The 5-year OS was 34%. The reduction in immunosuppression (*p* = 0.026) and number of ECP sessions (*p* < 0.001) were associated with improved OS, irrespective of other possible confounder factors. In particular, optimal OS was observed in patients who received more than 19 ECP sessions (*p* < 0.001).

### 2.2. ECP for Patients with cGvHD

#### 2.2.1. Clinical Characteristics of Patients with cGvHD

Regarding chronic GvHD patients, we studied 112 patients with moderate or severe chronic GvHD. Of the 112 patients initially evaluated to participate in the study, only 13 patients received four or fewer ECP sessions because of severe GvHD-related morbidity and were excluded from further analysis. A total of 99 patients (65 males and 34 females) with a mean age of 38.3 ± 13.2 years were included in our final analysis. The baseline characteristics of cGvHD patients who were treated with ECP are presented in Table 2.

Of the 99 patients that eventually completed the prespecified protocol procedures, 67 patients (67.7%) presented at baseline with cutaneous sclerosis manifestations, 73 patients (73.7%) with mucocutaneous disease, 36 patients (36.4%) with liver involvement, 42 patients (42.4%) with visceral manifestations, and 27 patients (27.3%) with lung involvement, as presented in Figure 2.

#### 2.2.2. Safety and Efficacy of ECP for cGvHD

Over the last two decades, ECP was commenced as a second-line treatment in 35 patients (35.4%), while the others received two previous lines of therapy. Ruxolitinib was administered in combination with ECP in 19 patients (19.2%) and with ibrutinib in 2 (2%) (see Table 2). The median number of ECP sessions was 17 (IQR: 6–49), while no serious ECP-related adverse events were observed. During the follow-up period, bacterial infections were observed in 43 patients (43.4%), viral in 38 (38.4%), and fungal in 11 patients (11.1%).

Only 19 patients (19.2%) did not show a response to ECP (Figure 3). Significantly lower rates of response were achieved in patients with visceral involvement (*p* = 0.037) and an earlier post-transplant GvHD diagnosis (*p* = 0.001).

Over a median follow-up period of 45.2 (IQR: 5.6–345.1) months, the 5-year CI of chronic GvHD-related mortality was 21.2% and was significantly reduced in patients who responded to ECP treatment (*p* < 0.001). Furthermore, the 5-year overall survival (OS) was 65.3% and was positively associated with better HLA matching (*p* = 0.011), a higher number of ECP sessions (*p* < 0.001), later post-transplant ECP initiation (*p* = 0.002), response to ECP (*p* = 0.036), and no relapse (*p* = 0.001).

## 3. Discussion

In this prospective study, based on real-world data, ECP was found to be safe and effective for aGvHD and cGvHD. It seems that overall survival for both aGvHD and cGvHD patients benefit from ECP and is associated independently with the number of sessions and the time of the treatment initiation. More specifically, the higher number of sessions and the earlier initiation inversely correlate with mortality, having a positive effect on the OS. In our study, there were no ECP-related major adverse events, but bacterial, viral, and fungal infections were present in patients who received combined ECP treatment for both acute and chronic GvHD.

We reported that 67.9% of our patients with aGvHD responded to ECP, and overall survival was associated with the number of ECP sessions. Similarly to our results, Kaya et al. found that increased an OS was associated with a prolonged duration of ECP therapy [22]. In their study, Asensi Cantó and colleagues showed that the 1-year OS of patients with aGvHD was significantly higher in those who responded to ECP compared to those who did not (*p* = 0.037) [23]. In their multicenter study including 75 allo-HCT recipients who developed aGvHD and were treated with ECP, Batgi et al. reported that the overall response rate was 48.6% [24]. In a meta-analysis of 7 studies and 121 patients, the overall response rate was 71% [25]. The differences between the response rates among the published studies might be attributed to the different criteria and various disease sites of gastrointestinal (GI) serious involvement used by the researchers for the evaluation of treatment response. Sohl et al. have shown that OS was significantly higher in aGvHD patients who received ECP compared to those who did not [26].

In our study, 80.8% of the patients with cGvHD responded to ECP. In the real-life study of Michallet et al., the response rate of ECP in patients with cGvHD was 78%, similar to our findings [27]. Linn et al. studied 75 alloHSCT recipients with cGvHD and found that the 12-month response rate was 70%, and treatment failure was associated with ECP resistance and relapse of malignancy [28]. The 5-year OS of our patients’ cohort treated with ECP was 65.3%. In the systematic review and meta-analysis of DeFilipp et al., the 5-year OS was 57.96% (data from eight studies and 431 patients) [29]. Compared to our data, similar response rates to ECP have been reported in children with GvHD [30]. Moreover, ECP has been found to be a highly cost-effective option for the treatment of cGvHD, while improving the quality of life of these patients [31,32]. Ali et al. conducted a randomized controlled trial, examining the prophylactic use of ECP in allo-HCT recipients [33]. However, they failed to show the benefit of ECP as an adjunct to standard drug-based GVHD prophylaxis [33]. Recently, ECP has been found to induce the formation of neutrophil extracellular traps in cGvHD patients [34]. Future research approaches should focus on the molecular and biochemical mechanisms implicated in the therapeutic effect of ECP.

Early in the course of the disease, it is anticipated that a combination of ECP with novel agents will benefit more patients with chronic or acute GvHD. Various novel agents have been approved by the Food and Drug Administration (FDA) for the treatment of steroid refractory (SR) GvHD. Ruxolitinib is a selective Janus kinase (JAK 1/2) inhibitor that blocks the JAK signaling pathway, preventing cytokine-driven tissue damage [35]. The FDA approval for SR-GvHD was based on the results of the REACH1 trial for SR-aGvHD and the REACH3 trial for SR-cGvHD [36,37]. Ibrutinib is an oral selective, irreversible inhibitor of Bruton’s tyrosine kinase (BTK), which was FDA-approved for SR-cGvHD after the PYC-1129 trial [38]. Ibrutinib prevents signal transduction and the activation of B-cells and interleukin-2-inducible T-cell kinases (ITK) [39]. Belumosudil is an oral selective inhibitor of rho-associated coiled-coil-containing protein kinase-2 (ROCK2) that was FDA-approved for SR-cGvHD after the results of the ROCKstar trial [40]. Belumosudil reduces Th17 and follicular helper cells and enhances regulatory T-cells, while at the same time, it downregulates profibrotic gene expression, as has been shown by both experimental and translational data [41,42]. This ability to simultaneously target both inflammation and fibrosis makes it a unique agent in the treatment of cGvHD. Recently, Axatilimab-csfr, a colony-stimulating factor-1 receptor-blocking antibody, was approved by the FDA for the management of patients with cGvHD [43]. Several other agents, including immunosuppressive drugs, monoclonal antibodies, and agents with endothelial protective properties are under investigation for the management of SR GvHD [44]. More prospective clinical trials on the efficacy of combined treatment of GvHD with these novel agents and ECP are essential [45].

Recently published studies have compared the efficacy of ECP versus Ruxolitinib regarding the treatment of SR-cGvHD. The results from a retrospective study by the EBMT Transplant Complications Working Party showed that there was not a statistically significant difference in OS, progression-free survival, non-relapse mortality, and relapse incidence between patients who received ECP and those who were treated with Ruxolitinib [46]. An interesting outcome of this particular study is that there were no major differences regarding the type and frequency of infections between the two treatment cohorts, which is surprising since there are theoretical benefits of ECP regarding the infection risk as compared with immunosuppressive therapies, such as Ruxolitinib [46]. Similarly, a difference in response rates and OS was not found in aGvHD patients who were treated with ECP and those who received Ruxolitinib [47]. Furthermore, ECP combined with Ruxolitinib might have potentially complementary mechanisms of action and their combination should be investigated for GvHD treatment [20].

Although these novels agents have shifted the therapeutic landscape toward more selective targets, additional investigation is needed in order to further enhance the treatment of GvHD. It is also important to highlight that even in the era of these novel biologics, ECP should be considered early in the course of GVHD, before significant irreversible end-organ damage has been established. It is difficult to evaluate the treatment of novel agents, corticosteroids, or ECP as a monotherapy since it is very rare to achieve a response in severe aGvHD or cGvHD with only one modality as pathogenetic mechanisms are very complex.

Some limitations should be acknowledged in our study. Firstly, it was a single-center study based on real-world data. Moreover, a comparison was not performed between ECP and other treatment options for GvHD. The potential biases of the study would be eliminated by including a larger number of patients. The assessment of the best response evaluation was performed at the end of all ECP treatment sessions. Another limitation that has to be recognized is that our work started before the establishment of the National Institutes of Health (NIH) 2015 response criteria for cGVHD and Magic criteria for aGVHD, and therefore, we did not perform retrospective evaluations against them [48,49,50].

## 4. Materials and Methods

### 4.1. Study Desing and Population

This is a prospective single-center study, in which patients who underwent allo-HCT at the JACIE accredited center of the Hematology clinic of G. Papanicolaou Hospital were included. Allo-HCT was performed in adherence to the indications established by the EBMT [51]. The study was conducted in accordance with the Declaration of Helsinki and was approved by the Ethics committee of G. Papanicolaou Hospital. All patients gave informed consent in order to be included in the study.

We enrolled consecutive adult patients who received ECP post-allo-HCT over the last 2 decades (2003–2022) for cGvHD and over the last decade (January 2013–December 2022) for steroid-dependent or refractory grade II-IV aGvHD. In every patient, the baseline evaluation including a clinical examination, blood counts, biochemistry values, clinical diagnosis of GvHD, and biopsy of the affected organs. Patients’ clinical and laboratory data were reported in real time and in detail. In these data, patients’ age, gender, hematological disease, conditioning regimen, donor type, disease risk index at diagnosis, occurrence of infections, clinical manifestations and grade of GvHD, prior treatments for GvHD, and concomitant treatments were included.

### 4.2. GvHD Grading and Prophylaxis, and ECP Protocol

AGvHD was diagnosed and graded based on the criteria of Glucksberg et al., while cGvHD assessment and grading were performed according to the National Health Institute criteria [51,52,53,54]. GvHD prophylaxis included cyclosporine–MTX in myeloablative and cyclosporine–mycophenolate mofetil (MMF) in reduced toxicity/intensity regimens. All patients with unrelated and some haploidentical donors received thymoglobulin (ATG) 5 mg/kg as prophylaxis.

Steroid refractoriness for aGVHD was defined by any of the following: progression of aGvHD after 3 days of treatment with methylprednisolone ≥2 mg/kg/day; or lack of response after 7 days of treatment [55]. Steroid-dependent aGvHD patients were considered those in whom the dose of methylprednisolone could not be reduced to 0.5 mg/kg/day or of prednisone to <0.6 mg/kg/day. Furthermore, steroid refractoriness for cGVHD was defined by any of the following: progression of cGvHD after 2 weeks of treatment with prednisolone ≥ 1 mg/kg/day; or the requirement for prednisone dose at least 0.5 mg/kg/day for a minimum of 4 weeks [56]. Steroid-dependent cGvHD patients were considered those in whom the long-term dose of prednisolone could not be reduced to 0.25 mg/kg/day (or 0.5 mg/kg/every other day) or less.

Before Ruxolitinib availability, ATG was commenced simultaneously with ECP initiation in steroid-refractory or dependent aGVHD patients. For patients with aGvHD, ECP was administered as a second-line therapy, based on the following protocol: two sessions every week for 1 month, one session every 2 weeks for 3 months, and one session every month for 6 months. Before Ruxolitinib or ibrutinib availability, MMF, cyclosporine, or ATG was commenced as a second-line treatment in steroid-refractory or dependent cGvHD patients, depending on previous prophylaxis. ECP was mostly administered as a third-line treatment for cGVHD. ECP was performed for cGvHD according to the following protocol: 1 session/week for the 1st month, 1 session/2 weeks for 3 months, and 1 session/month for 6 months, as reported in our study and in the National Comprehensive Cancer Network (NCCN) guidelines. Therakos, CELLEX machine was used for ECP. Specifically, CELLEX is a closed system in which leukapheresis, photoactivation with 8-MOP/UVA (methoxypsoralen; Uvadex), and reinfusion to the patient take place. Heparin was used to anticoagulate the circuit. The assessment of the best response evaluation was performed at the end of all ECP treatment sessions.

The criteria of ECP response were the following [57,58]:Complete response: absence of GvHD manifestations for both cGvHD and aGvHD;Very good response: for only aGvHD, grade I;Partial response: more than 50% in terms of organ involvement for both cGVHD and aGVHD.

### 4.3. Statistical Analysis

The primary statistical endpoint was the overall response rate to ECP treatment. The secondary endpoints included survival outcomes (OS), as well as infectious and other complications. A statistical analysis was performed using SPSS. 28.0 (IBM SPSS Statistics for Windows, Version 28.0. Armonk, NY, USA: IBM Corp). Descriptive statistics of categorical variables were presented as frequencies. Continuous variables for normality were assessed for normality, while normal was expressed as mean and non-normal as median and IQR. The Kaplan–Meier method was performed for survival analysis and a long-rank test was used for the comparisons between the different groups. A Cox regression model was used in the multivariate analysis for survival. Variables included in the analysis were: patient (age, disease) and transplant (donor, graft) characteristics and post-transplant outcomes (infections, GVHD, treatment, response to ECP, reduction in immunosuppression, time of ECP initiation, number of ECP sessions, relapse of malignancy, OS). Optimal binning was used to determine the optimal number of ECP sessions cutoff for statistical differences in overall survival. The statistical significance level was set at *p* < 0.05.

## 5. Conclusions

Our data confirm that ECP is a safe and effective treatment option for patients with acute and chronic GVHD and it is associated with an increase in overall survival for both types of GvHD. The option of monotherapy is very rare and indeed should be one goal of successful ECP. Although ECP is considered as a better tolerated, non-immunosuppressive treatment for GvHD, it remains unclear whether its effectiveness could be increased with novel treatments. In addition, the use of ECP in the treatment of patients with GvHD is not based on large prospective studies and specific therapeutic algorithms, indicating that this therapeutic option remains patient individualized among other GvHD treatments. In conclusion, further studies are needed to explore the possible effects of ECP-novel biologic combination treatments in patients with GvHD.

## Figures and Tables

**Figure 1 pharmaceuticals-17-01279-f001:**
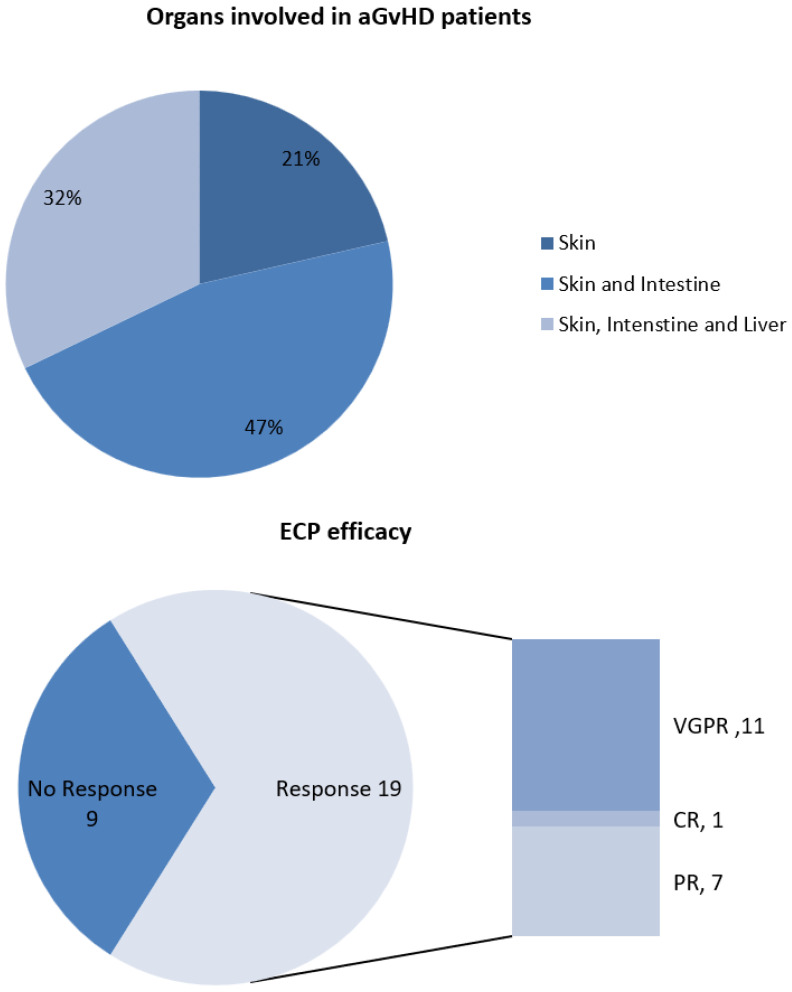
(**Above**) Clinical manifestations of patients with aGvHD, (**Below**) Efficacy of ECP in patients with GvHD. aGvHD = acute graft-versus-host disease; ECP = extracorporeal photopheresis; VGPR = very good response; CR = complete response; PR = partial response. Intestine stands for lower gastrointestinal tract.

**Figure 2 pharmaceuticals-17-01279-f002:**
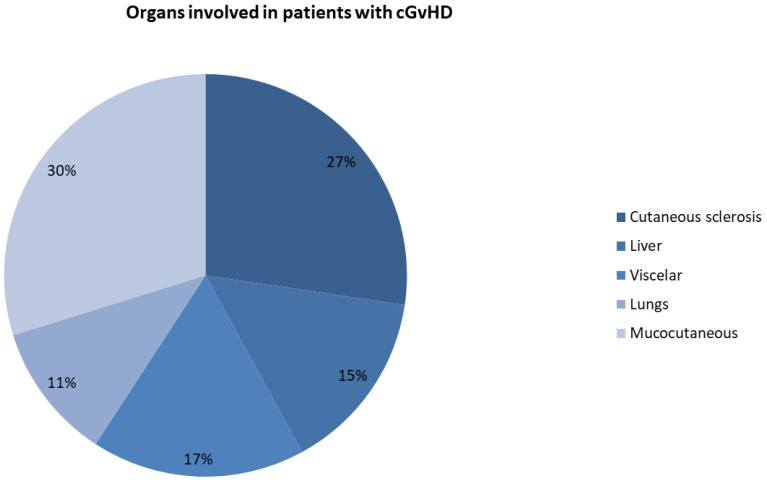
Clinical manifestations of patients with cGvHD who were treated with ECP. cGvHD = chronic graft-versus-host disease; ECP = extracorporeal photopheresis.

**Figure 3 pharmaceuticals-17-01279-f003:**
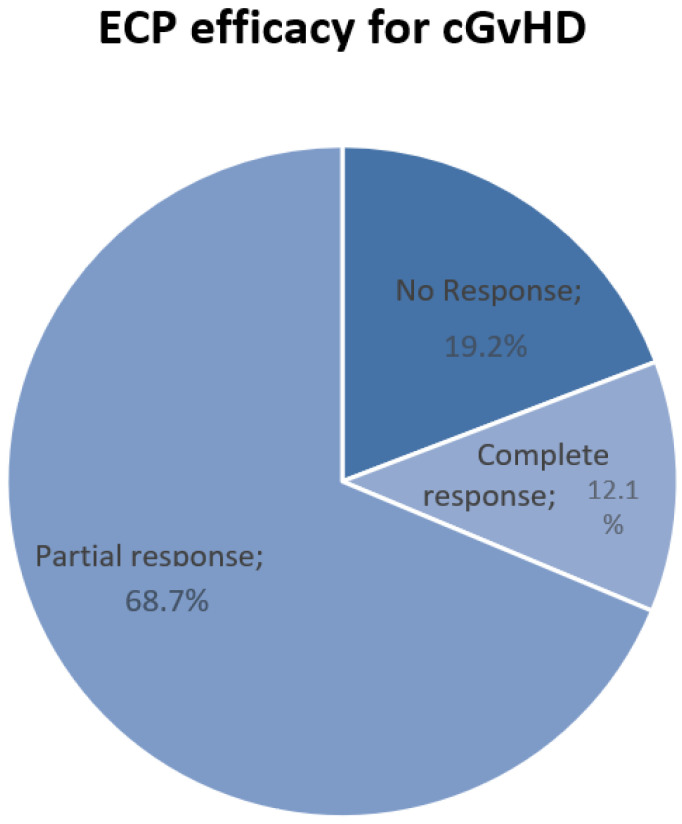
ECP efficacy for patients cGvHD. ECP = extracorporeal photopheresis; cGvHD = chronic graft-versus-host disease.

**Table 1 pharmaceuticals-17-01279-t001:** Clinical characteristics of patients with aGvHD, who were treated with ECP as second-line treatment.

Parameter	Study Population (N = 28)
Gender (*n*, %)	
Male	12 (42.9%)
Female	16 (57.1%)
Mean Age (years)	44.3 ± 15
Disease (*n*, %)	
AML	10 (35.7%)
ALL	8 (28.6%)
MDS	5 (17.9%)
MPN	1 (3.6%)
NHL	3 (10.7%)
HL	1 (3.6%)
Conditioning regimen (*n*, %)	
Myeloablative	16 (57.1%)
Reduced toxicity	8 (28.6%)
Reduced intensity	4 (14.3%)
Donor type (*n*,%)	
Sibling	4 (14.3%)
Matched unrelated	11 (39.3%)
One locus mismatched unrelated	12 (42.9%)
Haploidentical	1 (3.6%)
Disease Risk Index (*n*, %)	
Very high	1 (3.6%)
High	11 (39.3%)
Intermediate	14 (50%)
Low	2 (7.1%)
Median (IQR) days of aGVHD onset	17 (8–50)
Response to first-line treatment with corticosteroids	
Steroid-dependent	13 (46.4%)
Steroid-refractory	15 (53.6%)
Concomitant treatments (*n*, %) Steroids in tapering Cyclosporine	28 (100%) 17/28 (60.7%)
Median (IQR) days of ECP commencement post-HCT	18 (8–56)

aGvHD = acute graft-versus-host disease; ECP = extracorporeal photopheresis; AML = acute myeloid leukemia; ALL = acute lymphocytic leukemia; MDS = myelodysplastic syndrome; MPN = myeloproliferative neoplasm; NHL = non-Hodgkin lymphoma; HL = Hodgkin lymphoma; IQR = interquartile range; HCT = Hematopoietic cell transplantation.

**Table 2 pharmaceuticals-17-01279-t002:** Clinical characteristics of patients with cGvHD, who were treated with ECP.

Parameter	Study Population (N = 99)
Gender (*n*, %)	
Male	65 (65.7%)
Female	34 (34.3%)
Mean Age (years)	38.3 ± 13.2
Disease (*n*, %)	
AML	37 (37.4%)
ALL	28 (28.3%)
MDS	3 (3%)
MPN	6 (6.1%)
NHL	16 (16.2%)
HL	6 (6.1%)
MM	3 (3%)
Conditioning regimen (*n*, %)	
Myeloablative	70 (70.7%)
Reduced toxicity	25 (25.3%)
Reduced intensity	4 (4%)
Response to corticosteroids Steroid-dependent Steroid-refractory	38 (38.4%) 24 (24.3%)
ECP line (*n*, %)	
Second-line treatment	35 (35.4%)
≥Third-line treatment	64 (64.6%)
Concomitant treatments (*n*, %)	
Steroids in tapering Mycophenolate mofetil Cyclosporine Ruxolitinib	64 (65%) 31 (32%) 25 (26%) 19 (19.2%)
Ibrutinib	2 (2%)
ECP efficacy Complete response Partial response No response	12 (12.1%) 68 (68.7%) 19 (19.2%)

cGvHD = chronic graft-versus-host disease; ECP = extracorporeal photopheresis; AML = acute myeloid leukemia; ALL = acute lymphocytic leukemia; MDS = myelodysplastic syndrome; MPN = myeloproliferative neoplasm; NHL = non-Hodgkin lymphoma; HL = Hodgkin lymphoma; MM = multiple myeloma.

## Data Availability

Data is contained within the article.

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
