# Peer review of "Safety and Efficacy of Extracorporeal Photopheresis for Acute and Chronic Graft-versus-Host Disease"

_pharmaceuticals, 2024, doi:10.3390/ph17101279_

Round 1

Reviewer 1 Report

Comments and Suggestions for Authors

I read with attention and interest the contribution presented by Eleni Gavriilaki and coworkers entitled:

Safety and Efficacy of Extracorporeal Photopheresis for Acute and Chronic Graft-versus-Host-Disease

Manuscript ID: pharmaceuticals-3227918

The authors present their twenty-year experience in the treatment of Graft versus Host Disease with extracorporeal photopheresis. This paper reports a single centre experience gained in a Bone Marrow Transplant unit single accredited by JACIE. In the materials and methods section it is explained that patients were enrolled prospectively while data were extracted retrospectively from clinical documentation. The authors distinguish, correctly, in a very clear manner patients with acute GvHD from patients with chronic GvHD.

The study seemed to me to be well conducted overall.

The Introduction section appears verbose to be of adequate length and adequately and completely presents the rationale of the study. It is believed that some lines could be eliminated gaining in conciseness without losing clarity.

Lines 48-50

Lines 82-88

Moreover the meaning of the acronym mTOR must be explained the first time it is used (line 60)

The materials and methods section in my opinion should be integrated with a subsection that describes the methodology with which the ECP procedures were carried out. Did the authors use an off-line system or an in-line system? What equipment did they use (it might be sufficient to indicate the brand and model)? Were the cell separators and UVA irradiation devices used according to the standard settings or were they customized?

The results section appears to be understandable. The results are described in the text and are extensively presented in the form of tables and figures that simplify their understanding. The same data are not presented redundantly in tables and figures. Personally, I do not understand the usefulness of the figure in which the duration of survival is related to the number of FCE procedures performed. The FCE is suspended, for example, in the case of recurrence of haematological disease, sepsis, serious impairment of the patient's general condition. Therefore, it seems logical to believe that patients who must discontinue the ECP have a worse prognosis. Perhaps the figure could be eliminated, gaining in conciseness without losing clarity.

The discussion section appears adequate in length and content. The results obtained are summarized and discussed in light of the bibliography.

The conclusions section also appears adequate

References appears to be adequate and up to date.

Author Response

Dear Reviewer,

I read with attention and interest the contribution presented by Eleni Gavriilaki and coworkers entitled:

Safety and Efficacy of Extracorporeal Photopheresis for Acute and Chronic Graft-versus-Host-Disease

Manuscript ID: pharmaceuticals-3227918

The authors present their twenty-year experience in the treatment of Graft versus Host Disease with extracorporeal photopheresis. This paper reports a single centre experience gained in a Bone Marrow Transplant unit single accredited by JACIE. In the materials and methods section it is explained that patients were enrolled prospectively while data were extracted retrospectively from clinical documentation. The authors distinguish, correctly, in a very clear manner patients with acute GvHD from patients with chronic GvHD.

The study seemed to me to be well conducted overall.

Answer: We would like to thank the reviewer for the time dedicated to reviewing our work. Thanks for the positive feedback and comments.

The Introduction section appears verbose to be of adequate length and adequately and completely presents the rationale of the study. It is believed that some lines could be eliminated gaining in conciseness without losing clarity.

Lines 48-50

Lines 82-88

Answer: Thanks for this idea. These lines were removed as you suggested.  

Moreover the meaning of the acronym mTOR must be explained the first time it is used (line 60)

Answer: We are sorry for this omission. The acronym mTOR is now explained in the revised version of the manuscript.  

The materials and methods section in my opinion should be integrated with a subsection that describes the methodology with which the ECP procedures were carried out. Did the authors use an off-line system or an in-line system? What equipment did they use (it might be sufficient to indicate the brand and model)? Were the cell separators and UVA irradiation devices used according to the standard settings or were they customized?

Answer: We would like to thank the reviewer for this interesting suggestion. Please find in the materials and methods section the information asked about the ECP procedure.  

The results section appears to be understandable. The results are described in the text and are extensively presented in the form of tables and figures that simplify their understanding. The same data are not presented redundantly in tables and figures. Personally, I do not understand the usefulness of the figure in which the duration of survival is related to the number of FCE procedures performed. The FCE is suspended, for example, in the case of recurrence of haematological disease, sepsis, serious impairment of the patient's general condition. Therefore, it seems logical to believe that patients who must discontinue the ECP have a worse prognosis. Perhaps the figure could be eliminated, gaining in conciseness without losing clarity.

Answer: Thanks for the suggestion. Figure 2 was removed as the reviewer suggested.  

The discussion section appears adequate in length and content. The results obtained are summarized and discussed in light of the bibliography.

The conclusions section also appears adequate

References appears to be adequate and up to date.

Answer: We would like to thank the reviewer for the positive comments. Thanks once again.

Reviewer 2 Report

Comments and Suggestions for Authors

Dear Authors,

your work is very interesting and full of ideas to explore further. There are few real-world studies in the literature on ECP and even fewer that can evaluate combination therapy with the new recent drugs. I propose the following observations, many of which are dictated by the great curiosity aroused, which aim to highlight some interesting aspects of the work.Besides a few minor corrections

ROW 20: correct “graft-versus-host disease” in “graft-versus-host-disease”

ROW 24: indicate the population included (aGVHD steroid-refractory or dipendenti, and cGVHD > 2 or 3 line).

ROW 60: could the platform with PTCy also be indicated for completeness?

ROW 75: The author says that there are no standard second-line treatments, however the FDA and EMA have approved Ruxolitinib as a second-line therapy in aGVHD and cGVHD

ROW 75: correct “It is critical for both acute and chronic GVHD…”

ROW 77: The author indicates ECP as the first choice, but in light of the approval of Ruxolitinib, it would be better to indicate “one of the first choices”.

LINE 88: Although still debated, some studies show a contribution of ECP in inducing an increase in Tregs (which could explain the modulation of immunotolerance associated with the procedure)

Also indicate the steroid-sparing advantage of ECP (for example Novitzky-Basso 2023, Greinix 2022)

Insert Materials and Methods after the introduction, and then the results.

ROW 304: indicate the definition of steroid dependence and steroid refractoriness for acute and chronic GVHD

ROW 305 and 309: only steroid-refractory is indicated; and in the case of steroid-dependence?

ROW 311: The authors indicate the assessment of response at 5 days; this is the timing in acute GVHD, generally; in chronic, the assessment of the response of the first line is at least after 30 days (Penack 2020; Schoemans HM, EBMT-NIH-CIBMTR Task Force position statement on standardized terminology & guidance for graft-versus-host disease assessment. Bone Marrow Transplant. 2018). Can you explain the timing of this evaluation better?

ROW 312-313: acute GVHD is indicated in the ECP protocol for cGVHD

ROW 168, indicated the use of ECP also in the second line; indicate in the methods when ECP was used as second line and the type of protocol, if different

If possible, describe the characteristics of ECP used (inline or offline, volume processed,…)

ROW 316: The works cited do not take into account the new criteria for the evaluation of the NIH 2015 response for cGVHD or Magic criteria for aGVHD; furthermore, VGPR is not indicated in any of the works cited; specify the rationale for not using the Magic evaluation system for aGVHD and NIH 2014 for cGVHD. If you want to adopt an ECP-specific evaluation, integrate it with no response, mixed response, stable disease and progression (for cGVHD)

The assessment of the response evaluation must be better specified in both acute and chronic GVHD: are specific timings considered or is the best response obtained considered, during or at the end of the treatment? 

LINE 319: better “for only aGVHD”

LINE 320: better “for both cGVHD and aGVHD”

From line 120, the table could be integrated by indicating the timing of the onset of acute GVHD, the grading at the time of onset and the start of ECP

Indicate concomitant immunosuppressive therapy during ECP

In figure 1: lost the “R” of Liver; does “intestine” stand for upper and lower gastrointestinal tract or just for lower gastrointestinal tract?

ROW 135: It indicates "ECP was commenced at median day +18." When do you start counting the days? From the steroid or the onset of GVHD?

ROW 139: which immunosuppressive therapy was in progress? Was it a treatment that had already started during the transplant or added as a second line along with ECP? 

ROW 141: when did the infections occur? Specify if there were life-threatening infections and if the infections occurred during ECP and immunosuppressive therapy at full or reduced dosage.

ROW 145: what are the possible confounder factors?

A table with the variables evaluated would be useful.

Specify the occurrence or absence of serious (non-infectious) adverse events during ECP

Indicate in Table 2 as in Table 1, steroid-refractory, dependent, and steroid-untreated patients

Indicate how many patients had had previous GVHDa and how many had been treated with ECP; are there overlaps?

Indicate in Table 2:

- Patients divided by a moderate or severe form

- grading and staging organ by organ at baseline (to be able to see the response organ by organ, if possible)

- Timing of onset of cGVHD compared to transplant

- Timing of ECP from transplant and the previous line

LINE 171: when did the infections occur? Specify whether it is during or after ECP and if there is a difference in patients with concomitant therapy.

Unlike aGVHD, the type of response obtained with ECP in cGVHD is not clear; a graph like for aGVHD would be useful

It would be interesting to indicate the responses also organ by organ.

Is there a difference in response between forms of cGVHD with concomitant therapy? Did the use of ECP allow you to stop taking other medications?

LINE 175: is there a cut-off to identify the term “earlier” post-transplant?

LINE 180: as for aGVHD, the table with the variables and relative cut-offs is useful

Since the blood tests were considered, is there a correlation between lymphocytes at the start of ECP and response to ECP?

Among the discussions, line 261 talks about the studies comparing Ruxo and ECP; it would be useful to mention the work of Greinix on the combined and synergistic use of Ruxolitinib + ECP

Author Response

Dear Authors,

your work is very interesting and full of ideas to explore further. There are few real-world studies in the literature on ECP and even fewer that can evaluate combination therapy with the new recent drugs. I propose the following observations, many of which are dictated by the great curiosity aroused, which aim to highlight some interesting aspects of the work. Besides a few minor corrections

Answer: We would like to thank the reviewer for the time dedicated to reviewing our manuscript and the positive comments and feedback. The reviewer’s comments and corrections were indeed valuable for the quality and clarity of our work.  

ROW 20: correct “graft-versus-host disease” in “graft-versus-host-disease”

Answer: We would like to thank the reviewer for the correction. The phrase was corrected.  

ROW 24: indicate the population included (aGVHD steroid-refractory or dipendenti, and cGVHD > 2 or 3 line).

Answer: Thanks for the suggestion. We added the information that you asked.

ROW 60: could the platform with PTCy also be indicated for completeness?

Answer: We would like to thank the reviewer for the comment. The data for post-transplant cyclophosphamide that you asked for was added.

ROW 75: The author says that there are no standard second-line treatments, however the FDA and EMA have approved Ruxolitinib as a second-line therapy in aGVHD and cGVHD

Answer: We are sorry for this statement. We corrected this sentence in the manuscript.

ROW 75: correct “It is critical for both acute and chronic GVHD…”

Answer: We updated the manuscript according to the suggestion of the reviewer.

ROW 77: The author indicates ECP as the first choice, but in light of the approval of Ruxolitinib, it would be better to indicate “one of the first choices”.

Answer: We would like to thank for this idea. We changed the manuscript accordingly.  

LINE 88: Although still debated, some studies show a contribution of ECP in inducing an increase in Tregs (which could explain the modulation of immunotolerance associated with the procedure)

Answer: Thanks for this interesting idea! We included your suggestion in the revised manuscript.

Also indicate the steroid-sparing advantage of ECP (for example Novitzky-Basso 2023, Greinix 2022)

Answer: The steroid-sparing advantage of ECP was indicated in the revised manuscript.

Insert Materials and Methods after the introduction, and then the results.

Answer: We totally understand your suggestion. However, according to the journal’s format Results and Discussion have to be presented before the Materials and Methods.

ROW 304: indicate the definition of steroid dependence and steroid refractoriness for acute and chronic GVHD

Answer: We would like to thank the reviewer for this idea. The definitions of steroid-refractory/ dependent acute and chronic GvHD were added in the materials and methods section.

ROW 305 and 309: only steroid-refractory is indicated; and in the case of steroid-dependence?

Answer: Thanks for the suggestion. We have revised our manuscript accordingly.

ROW 311: The authors indicate the assessment of response at 5 days; this is the timing in acute GVHD, generally; in chronic, the assessment of the response of the first line is at least after 30 days (Penack 2020; Schoemans HM, EBMT-NIH-CIBMTR Task Force position statement on standardized terminology & guidance for graft-versus-host disease assessment. Bone Marrow Transplant. 2018). Can you explain the timing of this evaluation better?

Answer: The reviewer is right. We focused on the need of early response assessment especially on acute GvHD.  Final assessment of response to ECP was made at the end of ECP sessions. We made the essential corrections in the revised version of our manuscript. The following phrase was added:

“The assessment of the best response evaluation was performed at the end of all ECP treatment sessions.”

ROW 312-313: acute GVHD is indicated in the ECP protocol for cGVHD

Answer: Thanks for this comment. We made the essential clarifications in the manuscript.

ROW 168, indicated the use of ECP also in the second line; indicate in the methods when ECP was used as second line and the type of protocol, if different

If possible, describe the characteristics of ECP used (inline or offline, volume processed,…)

Answer: Thanks for this recommendation. In the revised version of the manuscript, the ECP procedure that was followed is described.

ROW 316: The works cited do not take into account the new criteria for the evaluation of the NIH 2015 response for cGVHD or Magic criteria for aGVHD; furthermore, VGPR is not indicated in any of the works cited; specify the rationale for not using the Magic evaluation system for aGVHD and NIH 2014 for cGVHD. If you want to adopt an ECP-specific evaluation, integrate it with no response, mixed response, stable disease and progression (for cGVHD)

The assessment of the response evaluation must be better specified in both acute and chronic GVHD: are specific timings considered or is the best response obtained considered, during or at the end of the treatment?

Answer: We understand the concerns of the reviewer. The assessment of the best response evaluation was performed at the end of all ECP treatment sessions. Our work has started before the establishment of these criteria and therefore, we did not retrospectively evaluate them. We have added it in the discussion and limitation section.

LINE 319: better “for only aGVHD”

Answer: Thanks for your suggestion. The manuscript was updated accordingly.

LINE 320: better “for both cGVHD and aGVHD”

Answer: The manuscript was updated as you suggested.

From line 120, the table could be integrated by indicating the timing of the onset of acute GVHD, the grading at the time of onset and the start of ECP

Answer: Thanks for this suggestion. The data that you asked were added to Table 1. Our goal in acute GVHD was to start ECP as early as possible from onset. Therefore, grading was similar at onset and start of ECP.

Indicate concomitant immunosuppressive therapy during ECP

Answer: Concomitant therapy has been added in the Table as recommended.

In figure 1: lost the “R” of Liver; does “intestine” stand for upper and lower gastrointestinal tract or just for lower gastrointestinal tract?

Answer: We would like to thank for these corrections. Intestine stands for lower gastrointestinal tract.

ROW 135: It indicates "ECP was commenced at median day +18." When do you start counting the days? From the steroid or the onset of GVHD?

Answer: Thank you for giving us the opportunity to clarify this issue. Days count from transplant.

ROW 139: which immunosuppressive therapy was in progress? Was it a treatment that had already started during the transplant or added as a second line along with ECP?

Answer: Immunosuppressive treatment started as second line along with ECP. Concomitant treatments have been added in Table as requested.

ROW 141: when did the infections occur? Specify if there were life-threatening infections and if the infections occurred during ECP and immunosuppressive therapy at full or reduced dosage.

Answer: Infections were reported throughout the treatment period of ECP. There were no life-threatening infections.

ROW 145: what are the possible confounder factors? A table with the variables evaluated would be useful.

Answer: Variables have been added in methods

“Variables included in the analysis were: patient (age, disease) and transplant (donor, graft) characteristics and post-transplant outcomes (infections, GVHD, treatment, response to ECP, reduction in immunosuppression, time of ECP initiation, number of ECP sessions, relapse of malignancy, OS).”

Specify the occurrence or absence of serious (non-infectious) adverse events during ECP

Answer: There were no serious adverse events related to ECP.

Indicate in Table 2 as in Table 1, steroid-refractory, dependent, and steroid-untreated patients

Answer: Data have been added.

Indicate how many patients had had previous GVHDa and how many had been treated with ECP; are there overlaps?

Answer: There are no overlaps in ECP treatment. Patients treated with ECP due to aGVHD are included only in this group, and not in cGVHD. The essential changes have been made in the manuscript.

Indicate in Table 2:

- Patients divided by a moderate or severe form

- grading and staging organ by organ at baseline (to be able to see the response organ by organ, if possible)

- Timing of onset of cGVHD compared to transplant

- Timing of ECP from transplant and the previous line

Answer: In the revised version of the manuscript, Table 2 was updated with more data.

LINE 171: when did the infections occur? Specify whether it is during or after ECP and if there is a difference in patients with concomitant therapy.

Answer: Infections were reported throughout the treatment period of ECP. There were no life-threatening infections. We did not find a significant difference among different concomitant therapies.

Unlike aGVHD, the type of response obtained with ECP in cGVHD is not clear; a graph like for aGVHD would be useful

Answer: The reviewer is right. We added a figure as suggested by the reviewer.

It would be interesting to indicate the responses also organ by organ.

Is there a difference in response between forms of cGVHD with concomitant therapy? Did the use of ECP allow you to stop taking other medications?

Answer: We have added relevant data.

LINE 175: is there a cut-off to identify the term “earlier” post-transplant?

Answer: Time post-transplant was used as a continuous variable.

LINE 180: as for aGVHD, the table with the variables and relative cut-offs is useful

Answer: We have added variables in methods.

Since the blood tests were considered, is there a correlation between lymphocytes at the start of ECP and response to ECP?

Answer: The reviewer’s idea is very interesting. However, we failed to identify a relationship between lymphocytes at the start of ECP and response to ECP.

Among the discussions, line 261 talks about the studies comparing Ruxo and ECP; it would be useful to mention the work of Greinix on the combined and synergistic use of Ruxolitinib + ECP

Answer: Thanks for the idea! The discussion section was updated based on your suggestions.
